# Molecular Underpinnings of Brain Metastases

**DOI:** 10.3390/ijms26052307

**Published:** 2025-03-05

**Authors:** Maria A. Jacome, Qiong Wu, Jianan Chen, Zaynab Sidi Mohamed, Sepideh Mokhtari, Yolanda Piña, Arnold B. Etame

**Affiliations:** 1Department of Immunology, H. Lee Moffitt Cancer Center & Research Institute, Tampa, FL 33612, USA; maria.jacomellovera@moffitt.org; 2Department of Neuro-Oncology, H. Lee Moffitt Cancer Center & Research Institute, Tampa, FL 33612, USA; qiong.wu@moffitt.org (Q.W.); jianan.chen@moffitt.org (J.C.); sepideh.mokhtari@moffitt.org (S.M.); yolanda.pina@moffitt.org (Y.P.); 3School of Medicine, Tulane University, New Orleans, LA 70112, USA; zaynab@tulane.edu

**Keywords:** brain metastases, molecular mechanisms, genetic features, central nervous system, breast cancer, lung cancer, melanoma

## Abstract

Brain metastases are the most commonly diagnosed type of central nervous system tumor, yet the mechanisms of their occurrence are still widely unknown. Lung cancer, breast cancer, and melanoma are the most common etiologies, but renal and colorectal cancers have also been described as metastasizing to the brain. Regardless of their origin, there are common mechanisms for progression to all types of brain metastases, such as the creation of a suitable tumor microenvironment in the brain, priming of tumor cells, adaptations to survive spreading in lymphatic and blood vessels, and development of mechanisms to penetrate the blood–brain barrier. However, there are complex genetic and molecular interactions that are specific to every type of primary tumor, making the understanding of the metastatic progression of tumors to the brain a challenging field of study. In this review, we aim to summarize current knowledge on the pathophysiology of brain metastases, from specific genetic characteristics of commonly metastatic tumors to the molecular and cellular mechanisms involved in progression to the central nervous system. We also briefly discuss current challenges in targeted therapies for brain metastases and how there is still a gap in knowledge that needs to be overcome to improve patient outcomes.

## 1. Introduction

Brain metastases (BMs) are the most commonly diagnosed central nervous system (CNS) tumors in adults in the United States, estimated to occur up to ten times more frequently than primary brain tumors [1]. Advancements in treatments of primary tumors have led to prolonged survival of patients, increasing the pool of prevalent cancer patients at risk for BMs [2]. Coupled with advancements in neuroimaging and increased physician and patient awareness, diagnoses of BM can be made earlier, helping to improve patient outcomes. Lung cancer, breast cancer, and melanoma are widely recognized as the three most prevalent causes of BMs [3,4], with renal or gastrointestinal causes representing a good fraction of BMs in certain populations [1,5].

The challenge remains in characterizing the mechanisms under which BMs are initiated and how they progress. This encompasses exploring the molecular and genetic underpinnings of tumors linked to BMs, the factors influencing brain tropism, the dynamics between tumor cells and the brain’s microenvironment, as well as the key mechanisms driving therapy resistance. Developments in molecular science in recent decades have allowed researchers to obtain more information on the intrinsic metastatic progress of tumors but the brain remains an organ in which investigation of these fundamental molecular underpinnings has proved limited. In this review, we intend to compile the most up-to-date information and recent research made on metastatic mechanisms on the brain, focusing on the specifics of breast cancer, lung cancer, and melanoma in the hopes of uncovering research gaps that can be further investigated to improve targeted therapies and patient outcomes.

## 2. Brain Metastases Epidemiology

Despite their high frequency, there is a lack of systematic, nationwide reporting of BMs [2]. Although precise values of BMs are unknown, more than 100,000 people are diagnosed annually [2], and it has been estimated that around 20% of all patients with cancer will develop BMs [6]. However, only estimates can be made since incidence data of BMs from all cancer sites is not widely available, and most studies determine incidence, prevalence, and prognosis using data from the National Cancer Institute’s (NCI) Surveillance, Epidemiology, and End Results (SEER) system [7], which takes the presence of synchronous BM at time of diagnosis of primary tumor as its only metric [3,7].

Breast cancer, lung cancer, and melanoma are among the most commonly associated tumors with BMs [1,5]. Autopsy studies report the incidence of BMs from lung tumors to be as high as 52%, but this varies depending on the histology of the tumor, the patient’s sex, and the stage at the time of diagnosis [1]. For BMs from breast cancer, autopsy studies have reported an incidence of around 18–30%, while studies utilizing the SEER database usually report lower numbers [1,7,8]. This is largely due to the fact that BMs are rarely present at times of primary tumor diagnosis, excluding it from the database. Melanoma, ranked as the third leading cause of BMs, has shown incidence rates varying from 6% up to 28% in some population-based studies [3,8]. Research has demonstrated that BMs are more prevalent in melanomas located in the head and neck region compared to those on the extremities or trunk. This increased incidence is especially notable among male patients, younger individuals, and cases with greater Breslow thickness [9]. Further explorations about patient characteristics, such as age, sex, ethnicity, and their role in BM incidence, can be found in the work of Parker et al. [10].

The presence of synchronous BM at times of diagnosis of primary tumor is associated with poorer survival than finding extracranial metastases only, with median survival time being only 5 months [10]. A study comparing median overall survival (OS) in BMs patients from different solid tumors showed that breast cancer patients had the shorter OS, with 9.9 months in contrast to 10.3 and 13.7 months in melanoma and non-small cell lung cancer (NSCLC), respectively [11]. In contrast, another study suggested that with an OS of 10 months, breast cancer had longer survival than melanoma and NSCLC [3]. The molecular subtype of primary tumors significantly influences both the risk of BMs development and the associated prognosis [6]. In breast cancer, patients with human epidermal growth factor receptor 2 (HER2)-positive, hormone receptor-negative subtypes and those with triple-negative subtypes (negative for estrogen receptor (ER), progesterone receptor (PR), and normal HER2 expression) exhibit a higher incidence of BMs [12]. Notably, the triple-negative subgroup has the shortest median survival, at just 6.0 months [12]. In NSCLC, BMs are more frequently observed in cases with epidermal growth factor receptor (EGFR) mutations or anaplastic lymphoma receptor tyrosine kinase (ALK) rearrangements, with over 45% of patients developing CNS involvement within the first three years of survival [13]. Melanoma is the tumor with the strongest affinity for the CNS, with some autopsy studies reporting up to 75% of CNS involvement [14]. Jakob et al. found that CNS involvement was significantly higher in patients with mutations in the v-raf murine sarcoma viral oncogene homolog B1 (*BRAF*) and neuroblastoma RAS viral (v-ras) oncogene homolog (*NRAS*), occurring in 24% and 23% of cases, respectively [15]. In comparison, patients with wild-type genes exhibited a CNS involvement prevalence of only 12% [15].

Knowledge of the epidemiology of BMs has improved screening, diagnostic, and treatment standards, and these data have been used to create prognostication tools such as the recursive partitioning analysis (RPA) score [16] and the newer Graded Prognostic Assessment (GPA) [17]. GPA, in particular, integrates patient data such as age, Karnofsky Performance Score (KPS), number of BMs, and presence of extracranial metastases with histologic and molecular data from the primary tumor in a diagnosis-specific index (DS-GPA) [18]. An updated, user-friendly GPA calculator can be accessed at BrainMetGPA.com.

There is an imperative need for improvement in the reporting of BMs to help identify patients at the highest risk for BMs. Moreover, this could promote research into the mechanisms that make these patients develop BMs in the first place and develop targeted therapies.

## 3. Genetic Predisposition for Brain Metastases

Metastatic lesions have a different molecular and genomic landscape than their primary tumors [19]. Phylogenetic analysis of tumor variations and migration histories have shown that clonal branching can occur within the primary tumor as often as it occurs after egressing the primary tumor [20], making circulating tumor cells (CTCs) and circulating tumor DNA better resources for capturing the tumor heterogeneity of metastatic progression [21]. Genome-wide sequencing analyses of CTCs have identified some mutations that can be considered crucial for determining organ tropism. Brastianos et al. sequenced matched BMs, primary tumors, and normal tissue of 86 patients and observed branched evolution from a common ancestor in the metastatic lesions [19]. They found that extracranial and lymph node metastases diverged from BMs but that even spatially and temporally separated BMs were genetically homogeneous to each other, indicating that the genetic alterations acquired by brain-tropic tumor cells are different from those in other metastatic sites and probably grant an advantage for survival in the brain [19]. To identify metastatic signatures, researchers developed a MetMap using 500 cell lines from 21 different solid tumor types in mouse xenograft models. This map revealed patterns of metastasis that are specific to certain organs and linked these patterns to various clinical and genomic characteristics [22]. The MetMap can help determine common molecular and genetic alterations that enhance metastasis to certain organs and potentially find new therapeutic approaches. The complex evolution tumor cells must endure to reach the brain and survive in it requires the cooperation of genetic, epigenetic, transcriptomic, metabolic, and immunologic factors and will only occur in brain-tropic cells, prepared to go through all those changes. Figure 1 offers a summary of specific mutations associated with brain tropism in different primary cancer types.

### 3.1. Genetic Features of Breast Cancer Brain Metastases

Advancements in gene expression profiling have greatly enhanced our understanding of breast cancer. The expression levels of three pivotal receptors in breast cancer—estrogen receptor (ER), progesterone receptor (PR), and human epidermal growth factor receptor 2 (HER2/neu)—help categorize it into four primary molecular subtypes. These include luminal A (ER-positive, PR variable, HER2-negative), luminal B (ER-positive, PR variable, HER2-positive or negative), HER2-enriched (ER-negative, PR-negative, HER2-positive), and basal-like (ER-negative, PR-negative, HER2-negative). Notably, the basal-like subtype constitutes the majority of “triple-negative” breast cancers [23,24].

Breast cancer molecular subtypes have preferential sites for metastases and possess a protein profile associated with homing of the metastatic site. A large, registry-based, single-institution study showed that patients with the HER2-positive and triple-negative subtypes had the highest incidence of BMs [25]. This follows the line of previous reports showing that basal-like/triple-negative tumors pose a higher risk for BMs, with HER2-enriched in second place [26,27]. However, HER2 and hormonal receptor statuses are hypothesized to shift upon reaching the brain [26]. Similarly, triple-negative brain-metastatic cells exhibit elevated β2-adrenergic receptor mRNA and protein levels compared to their primary tumor counterparts, enhancing their proliferative capacity [28]. Triple-negative and basal-like breast cancer subtypes have been found to compromise the blood–brain barrier (BBB), whereas BMs from HER2/neu-positive breast cancers generally preserve BBB integrity [29]. This prompts the need for HER2/neu-positive cells to find alternative pathways to penetrate the BBB. HER2-HER3 dimers can form in breast cancer BMs and preferentially link to their ligand heregulin (also known as neuregulin-1) in the endothelial cells of the BBB [30]. Heregulin and HER2 signaling induces activation of extracellular cathepsin B and matrix metalloproteinase (MMP)-9, on which transmigration through the BBB is dependent [30]. MMP-9 enhances extracellular proteolysis and is upregulated by the metalloprotease-disintegrin *ADAM8*, which is highly expressed in all BMs but particularly breast cancer cells [31]. Cathepsins, on the other hand, are a family of proteases involved in protein degradation and processing, and their increased expression and activity could promote angiogenesis, invasion, and cell proliferation in some cancers [32,33]. In BMs, cathepsin S is produced by macrophages and breast cancer tumor cells, and it mediates BBB transmigration via proteolytic cleavage of the junctional adhesion molecule (JAM)-B [34]. Heregulin also upregulates intercellular adhesion molecule 1 (ICAM1), which is linked to increased invasion, motility, and metastasis in breast cancer [30]. Gene expression profiling of paired primary breast carcinomas and their corresponding BMs identified the upregulation of 1314 genes and the downregulation of 1702 genes in BMs relative to the primary tumors [35]. This study also showed activation of the HER2 pathway and gains in transcript and protein expression of rearranged during transfection (*RET*) gene in BMs, both linked to disease progression [35]. Interestingly, there was no loss in *PTEN* expression in the analyzed specimens, which has been reported as a driver for BMs induced by astrocytes [36]. A nationwide cohort study in Finland determined that basal-like subtypes tended to first metastasize to the brain and had a protein profile with high expression of neural cell stemness-linked proteins nestin and prominin-1, which could potentially help breast tumor cells adapt to the brain [37].

Jin et al. trialed their MetMap with breast cancer cells that metastasized to the brain and demonstrated these cells present an altered lipid metabolism, which is necessary for tumor cell survival within the brain microenvironment [22]. Increased fatty acid synthase (*FASN*) gene expression in breast cancer cells has been identified as a way to overcome low lipid availability in the brain [38]. Enzymes associated with glycolysis, the tricarboxylic acid cycle, and oxidative phosphorylation pathways show elevated expression and could further promote efficient energy production via glucose oxidation. Furthermore, the pentose phosphate pathway and the glutathione system demonstrate heightened activity, contributing to the reduction in reactive oxygen species [39]. Whether reprogramming occurs before seeding the brain or is induced by the lipid-poor environment of the brain is not known, but it is suggested that high *FASN* expression increases cell propensity to colonize the brain [38]. Increased levels of fatty acid binding protein 7 (FABP7) are also seen in HER2-positive breast cancer BMs, and besides its role in metabolic reprogramming, FABP7 upregulates metastatic genes and pathways, such as Integrins-Src, MEK/ERK, Wnt/β-catenin, and vascular endothelial growth factor (VEGF)-A [40,41]. Overexpression of proteins involved in fatty acid synthesis and degradation, as well as glucose-regulated protein 94 (GRP94), help cells compensate for the hypoglycemic stress they are subject to in the brain [42].

Other gene signatures expressed in breast cancer cells that metastasize to the brain include *COX2*, *HBEGF*, and *ST6GALNAC5*, which mediate BBB migration [43]; *PCDH7*, involved in linkage and interaction of tumor cells with astrocytes [44]; and *GRIN2B,* particularly increased in triple-negative breast cancers and involved in coding the GluN2 subunit of the NMDAR [45]. The glutamate-stimulated GluN2B-NMDAR signaling axis activation in cancer cells promotes colonization and metastatic tumor growth in the brain by forming pseudo-tripartite synapses in which tumor cells act as an astrocyte [45]. The *MYC* oncogene is highly expressed in CTCs from metastatic variants of breast cancer, and it regulates the adaptation of CTCs to the brain environment by reducing the oxidative stress produced by activated microglia via gene upregulation of glutathione peroxidase 1 (GPX1) [46]. A specific protein signature in CTCs consisting of HER2+/EGFR+/HPSE+/Notch1+ was termed “brain metastasis selected markers” by Zhang et al. and proved to increase CTC BMs compared with the parental CTC lines [47]. *SERPINA5* is significantly upregulated in breast cancer BMs, which induces the production of anti-PA serpins [46,48]. Also, the *GBP1* gene is upregulated in ER-negative breast cancer cells that develop BMs [49]. *GBP1* codes for the Guanylate-Binding Protein 1 (GBP1), binding activated T lymphocytes and enabling tumor cells to cross the BBB [49].

### 3.2. Genetic Features of Lung Cancer Brain Metastases

Lung cancer is primarily categorized into two major histological types: non-small cell lung cancer (NSCLC) and small cell lung cancer (SCLC) [50]. Within the NSCLC group, further classifications include adenocarcinoma, squamous cell carcinoma, and large cell carcinoma, among other subtypes [50]. Whole exome sequencing of samples from NSCLC and SCLC patients showed that NSCLC had a higher percentage of seemingly metastases-specific mutations, suggestive of branched evolution. In contrast, SCLC samples showed low heterogeneity, which suggests these tumors spread using a parallel and linear model of evolution [51]. In a systematic review of 72 studies comprising data from 2346 patients, the most common genetic alterations seen in BMs from NSCLC were *EGFR*, *TP53*, *KRAS* (Kirsten rat sarcoma viral oncogene), *CDKN2A* (cyclin-dependent kinase inhibitor 2A), and *STK11* [52]. Although not considered a driver gene, mutations in the tumor suppressor *TP53* are highly prevalent [53]. These mutations are linked to the development of new distant metastases [54] and exhibit strong concordance between primary NSCLC tumors and their corresponding BMs [55,56]. Mutations in the p53 protein disrupt cell cycle control, allowing the replication of damaged DNA and resulting in uncontrolled cell proliferation [57]. The EGFR family consists of four distinct members, all belonging to the ErbB/HER protein family: ErbB1, ErbB2, ErbB3, and ErbB4 [58]. Mutations in some tumors can continuously activate EGFR, enhancing tumor growth, invasion, and metastasis [59]. A meta-analysis of 26 studies demonstrated a positive association between *EGFR*-mutated NSCLC tumors and BMs, with an odds ratio (OR) of 1.58 (95% CI: 1.36–1.84), which confirms that *EGFR* mutation is a significant risk factor for BMs in NSCLC [60].

*ALK* gene fusion and *RET* gene fusion are also positive driver genes for NSCLC BMs [61]. *ALK* gene rearrangements frequently involve translocation or fusion with another partner gene, including echinoderm microtubule-associated protein-like 4 (*EML4*), the most prevalent in NSCLC [62,63]. *ALK* gene rearrangements result in the formation of an oncogenic ALK tyrosine kinase that persistently activates various downstream signaling pathways, including PI3K-AKT, MEKK2/3-MEK5-ERK5, JAK-STAT, and MAPK. This continuous activation promotes increased proliferation and survival of tumor cells [64]. *ALK* fusions have been found to be constant between primary NSCLCs and their associated BMs [65]. Compared to *EGFR*-positive groups, *ALK*-positive patients have a higher incidence of BMs at the time of initial lung cancer diagnosis [66]. However, Rangachari et al. found a similar baseline incidence of BMs in *EGFR*-mutated and *ALK*-rearranged NSCLCs and an evolutionary increase in CNS involvement over time, with >45% of patients in both groups showing BMs after three years of survival [13]. Larger BM tumor size has also been reported in the *EML4-ALK* fusion groups compared to groups without fusion [67].

The *RET* protooncogene codes for a receptor tyrosine kinase and has been identified in NSCLC rearranged or fused with over a dozen partner genes, with the kinesin family member 5B gene (*KIF5B*) being the most common [68]. *RET* fusion-positive NSCLCs have BMs in 25% to 50% of cases [69,70,71]. Although the exact mechanism of how *RET* fusion promotes brain organotropism for tumor cells is not known, recent trial results in patients receiving the selective RET inhibitor selpercatinib demonstrated decreased CNS metastatic progression of *RET* fusion-positive NSCLC, with no CNS involvement at all in patients with no previous BMs [72]. These results suggest *RET* plays a fundamental role in promoting tumor cell growth and survival in the brain.

The C-ros oncogene 1 (*ROS1*) encodes a receptor tyrosine kinase that is structurally analogous to *ALK* [73]. NSCLC harboring *ROS1* rearrangements exhibits a cumulative incidence of CNS metastasis comparable to that of *ALK* fusion-positive tumors [74]. The incidence of BMs in *ROS1*-rearranged NSCLC patients at the time of diagnosis is approximately 20–30%, while it is as high as 50% in patients post-crizotinib therapy [75]. Crizotinib, an ALK/MET kinase inhibitor developed for *ALK*-rearranged NSCLC, is also effective in treating *ROS1*-rearranged tumors [76]. However, it has low BBB penetration, and even with therapy, *ROS1*-positive patients commonly progress to CNS metastasis [77]. Recent clinical trials have demonstrated the promising efficacy of novel tyrosine kinase inhibitors (TKIs) in overcoming crizotinib resistance in BMs of *ROS1*-rearranged NSCLC [78,79]. Additionally, *MET* amplification in NSCLC leads to the heightened expression and continuous activation of the Met receptor, also known as the hepatocyte growth factor receptor (HGFR) [80]. This, in turn, promotes tumor cell migration and epithelial-to-mesenchymal transition phenotype [81]. Moreover, *MET* amplification has been found enriched in NSCLC BMs compared to paired primary tumors [82].

*KRAS* oncogene mutations are recognized as prevalent drivers in BMs from NSCLC, though their exact incidence varies across different studies [83,84]. The *RAS* genes encode a family of proteins that play critical roles in regulating cell growth, differentiation, and apoptosis [85], and *KRAS* mutation has been shown to upregulate PD-L1 expression in NSCLC through p-ERK signaling [86]. Activation of the PD-1/PD-L1 axis suppresses T-cell activity within the tumor microenvironment, allowing tumor cells to escape immune detection [87]. This immune regulation function of *KRAS*-mutations may improve tumor cell survival in the brain, but it also makes NSCLC BMs with *KRAS*-mutations more susceptible to treatment with immune checkpoint inhibitors (ICIs) [88].

Other less frequent mutations related to BMs from NSCLC include *BRAF* mutations [89], *Cav-1* [90], *AKT-1* [91], *NRAS*, and *PTEN* [92]. Gene expression signatures able to activate the WNT/TCF pathway are associated with lung adenocarcinoma metastases to the brain and lung [93]. The target genes *HOXB9* and *LEF1* within the WNT/TCF pathway play pivotal roles in facilitating chemotactic migration and promoting colony expansion in lung adenocarcinoma [93]. Furthermore, overexpression of the hyaluronan receptor by lung adenocarcinoma tumor cells increases inflammation and binding to hyaluronan-rich microenvironments such as the extracellular matrix of brain metastatic niches [94]. Aljohani et al. performed whole-genome sequencing on normal lung tissues, primary NSCLC tumors, their corresponding BMs, and CTCs. The study revealed that primary tumors contained mutations in genes associated with cell adhesion and motility. In contrast, BMs and CTCs exhibited mutations in genes responsible for adaptive and cytoprotective functions related to cellular stress responses, including *Keap-1*, *Nrf2*, and *P300* [95].

Several other adaptations can be seen in lung cancer CTCs. Analysis of tissue from lung adenocarcinoma and its matched CTCs and BMs using scRNA-seq have demonstrated that CTCs were in an intermediate place between BMs tumor cells, which leaned towards the epithelial phenotype and primary tumor cells, mostly found in a mesenchymal state [96]. Furthermore, *RAC1*, highly expressed in metastatic tumor tissue, was involved in adhesion, degradation, and VEGF signaling pathways [96]. CTCs overexpressing CD44v6 exhibited increased expression of the mesenchymal marker vimentin and reduced expression of the epithelial marker E-cadherin, thereby facilitating cell invasion and BMs through the activation of epithelial-to-mesenchymal transition [97].

### 3.3. Genetic Features of Melanoma Brain Metastases

The *BRAF* oncogene encodes a protein that is essential for the functioning of the mitogen-activated protein kinase/extracellular signal-regulated kinase (MAPK/ERK) signaling pathway [98]. *BRAF* gene mutations cause MAPK/ERK continuous activation and signal transduction, increasing cell growth, migration, and proliferation [98]. Approximately half of advanced melanomas harbor mutations in the *BRAF* gene [99], which are linked to an increased frequency of BMs [100,101]. Moreover, the incidence of *BRAF* mutations is higher in BMs than in primary melanomas or metastases to other organs, suggesting there may be an independent evolution of subclones [102].

V600E is the most frequently occurring *BRAF* mutation in melanoma, and it also has the highest association with BMs [99,101]. The loss of the PTEN protein has been shown to decrease the time to melanoma BMs in patients with *BRAFV600* mutations [103]. However, *BRAF* mutations alone are not sufficient for BMs to occur, and there is proof that *PTEN* gene silencing cooperates with *BRAFV600E* mutations in melanoma progression via the phosphoinositide 3-kinase (PI3K)/AKT pathway activation [104]. Although AKT1 activation can independently drive BM progression, it is augmented by *PTEN* silencing [105]. The bidirectional communication between the PI3K/AKT/mTOR and the MAPK/ERK pathways is critical for abnormal proliferation and therapy resistance in cancer [106].

BRAF-mutant melanomas have a significantly higher activation of AKT than *NRAS*-mutant melanomas [107]. However, *NRAS* mutations are also a significant risk factor for BMs, with a higher risk of developing BMs compared to patients with *NRAS* wildtype [15,108,109]. *NRAS* forms part of the Ras genes family, which also includes *HRAS* (Harvey Rat Sarcoma Virus) and *KRAS* (Kirsten Rat sarcoma virus) [110]. *NRAS* mutations are present in approximately 20% of human melanomas, whereas *HRAS* and *KRAS* mutations occur in only 1% and 2% of melanomas, respectively [111]. Both *NRAS* and *KRAS* mutations are enriched in melanoma BMs [112]. These mutated genes code for constitutively active Ras proteins that stimulate multiple signaling cascades, including the MAPK/ERK pathway and the PI3K/AKT pathway [110]. These are the same signaling cascades activated in tumors with *BRAF* mutations and *PTEN* silencing. However, *NRAS*-mutant melanomas exhibit normal PTEN levels, suggesting *BRAF*-mutant and *NRAS*-mutant tumors differ in their mechanisms of progression towards BMs [107,113]. Moreover, concurrent mutation of *NRAS* and *BRAF* is rare [113]. Evidence has also shown that the microenvironment in *NRAS*-mutant melanoma BMs is enriched in neutrophils in contrast to the primary melanoma [114]. In myeloproliferative neoplasms, a link between NRAS mutations and neutrophil hyperleukocytosis via activation of the granulocyte colony-stimulating factor (G-CSF) has been elucidated [115]. The exact mechanism of how neutrophils are seen in NRAS-mutated melanoma BMs remains to be exposed.

Other driver genes of melanoma are *NF1* inactivation and *C-KIT* mutations [116]. *NF1* functions as an inhibitor of Ras signaling, and its loss results in continuous activation of the MAPK and PI3K pathways. Likewise, mutations in *KIT*, a receptor tyrosine kinase, similarly initiate the activation of these pathways [117]. Patients with “quadruple negative” disease (no *BRAF*, *NRAS*, *NF1*, or *C-KIT* mutations) have the lowest risk of developing BMs [116]. This suggests the important implications of MAPK/ERK and PI3K/AKT activation in the metastatic progression of melanoma and their potential as therapeutic targets. Lessard et al. identified that metastatic melanoma cells exhibit elevated levels of the long intergenic non-coding RNA CASC15, which was associated with BMs in mouse xenograft models [118]. Additionally, increased expression of miR-301a in melanoma is linked to overall metastatic activity [119]. Although various other non-coding RNAs have been recognized as important regulators of melanoma progression and resistance to therapy, their specific roles in BMs remain unclear [120]. Alterations in the *CDKN2A* gene or the p16-cyclin D-CDK4/6-retinoblastoma protein pathway (CDK4 pathway) have been found in virtually all melanoma cell lines [121,122]. CDK4 activation inhibits the retinoblastoma protein, promoting cell cycle progression, and is usually associated with tumor suppressor *CDKN2A* (p16INK4A) deletion, furthering melanoma cell survival [121]. It has also been found that patients with deletions in *CDKN2A* genes also display *MDM2* and *MDM4* amplifications, which is associated with a higher risk for metastasis to the brain [123]. MDM2 and MDM4 are negative regulators of p53; therefore, amplifications in MDM2/4 decrease p53 function. The ubiquitin-specific protease 7 (USP7), a protein that protects MDM2/4 from proteasomal degradation, is increased in metastatic melanoma [124]. PPM1D, another negative regulator of p53, is overexpressed in metastatic melanoma, and gain-of-function mutations in immune cells promote immune escape and proliferation [125]. Even though *CDKN2A*, *MDM2/4*, *USP7,* or *PPM1D* are not considered driver mutations, their effect in the CDK4 and p53 pathways has a pro-metastatic effect.

The nerve growth factor (NGF) receptor CD271 is a low-affinity receptor for NGF, a member of the neurotrophin family proteins, highly expressed in melanoma cells even before BMs [126,127]. NGF and neurotrophin 3 (NT-3) are highly expressed in tumor-adjacent tissues in the brain, suggesting brain organotropism between the CD271-positive cells and the brain tumor niche [128]. CD271 has also been linked to SOX10, a specific marker of the neural crest, and it provides melanoma cells with neural crest stem cell signatures, a common ancestor between melanocytes, glial cells, and neurons [129]. It has been reported that aggressive melanoma cells hijack neural crest-related signaling pathways to increase plasticity and facilitate invasion in the brain [130]. *BMP4* and the Wnt target gene *AXIN2* are important for neural crest development and are also upregulated in BMs, suggesting melanoma acquires neuronal-like characteristics that make them highly efficient in metastasizing the brain [130]. Neurotrophins further stimulate invasion by producing extracellular matrix degradative enzymes such as heparinase, which destroys the basement membrane of the BBB [128]. AXL, a receptor tyrosine kinase, is involved in promoting epithelial-to-mesenchymal transition, treatment resistance, and metastasis in melanoma BMs [131]. AXL is typically upregulated in CD271-positive BMs and may actively contribute to immune escape. This occurs through mechanisms such as reduced HLA class I expression, increased production of immunosuppressive cytokines and PD-L1, and diminished infiltration of CD8+ lymphocytes [132]. The immune microenvironment of BMs has shown distinct characteristics when compared to their primary tumors or extracranial metastases, such as decreased IFNγ production and activated T-cells [114]. Moreover, there are fewer inflammatory cytokines, immune cell infiltrates, and maturation of dendritic cells, whereas there is enhanced oxidative phosphorylation [114]. Altogether, melanoma cells can invade and proliferate in the brain through a series of genetic, molecular, and immune mechanisms.

### 3.4. Genetic Signatures in Other Brain Metastases

The mechanisms for the metastatic spread of colorectal cancer to the brain are still not completely understood. There is an association between *RAS* mutations in colorectal tumors, especially *KRAS* mutations, and increased risk of BMs [133,134,135]. High expression of *NFAT5* (Nuclear Factor of Activated T Cells 5), *AVCR1C* (Activin A Receptor Type 1C), and/or *SMC3* (Structural Maintenance of Chromosomes 3) is associated with colorectal BMs [136]. Certain gene variants are also associated with increased risk for BMs and BBB penetration, such as *ST6GALNAC5*, which encodes for a sialyltransferase involved in cell–cell adhesion, and *ITGB3*, which encodes integrin β3, stimulating adhesion, migration, and angiogenesis [137]. While colorectal cancer was once believed to be an extremely rare cause of BMs, some studies have found that colorectal cancer patients can have an incidence of up to 14.6% of BMs [138]. Most of those patients were asymptomatic at the time of diagnosis. Furthermore, synchronous lung metastases increase the risk of BMs [139]. This finding suggests that the genomic or molecular alterations needed to metastasize to the brain are common to other sites and acquired early during metastatic progression.

A genomic next-generation sequencing study in renal cell carcinoma BMs found an enrichment of the *SMARCA4* gene in BM tumors in contrast with primary tumor and extracranial lesions [140]. *SMARCA4* encodes a subunit of the SWI/SNF chromatin-remodeling complex, which functions as an epigenetic regulator of gene expression and plays a critical role in tumor suppression [141]. Renal cell carcinoma that metastasized to the brain also showed more PI3K pathway alterations, primarily *PTEN* inactivation, than cells that did not metastasize to the brain [142]. As previously discussed with melanoma BMs, the PI3K pathway plays an important role in metastatic progression. Wyler et al. demonstrated that the expression of chemokines and their receptors play a significant role in the propensity of renal cell carcinoma to metastasize to the brain [143]. Specifically, the levels of the monocyte-specific chemokine CCL7 and its receptor CCR2 were found to be elevated in BMs compared to primary tumors [143]. This suggests that the recruitment of monocytes and macrophages is a key factor contributing to the establishment of BMs [143]. Of note, renal cell carcinoma is a highly immunogenic entity: Harter et al. demonstrated that they had the highest levels of CD3+ and CD8+ lymphocytes and the strongest PD-1 levels, which correlates to smaller brain tumor sizes [144].

## 4. Molecular Mechanisms of Metastatic Progression to the Central Nervous System

As early as 1889, Stephen Paget described the pattern in which breast cancers metastasized predominantly to certain organs more than others and compared it to seeds that can only grow on suitable soil [145]. Despite being over a hundred years old, this “Seed and Soil” theory continues to be valid today. The potential of tumor cells to metastasize depends on interactions between selected metastatic cells and mechanisms unique to some organ microenvironments in which chemotaxis and growth can occur [146]. Studies of multiphoton laser scanning microscopy have allowed us to follow individual tumor cells that metastasize to the brain and determine the multi-step process they go through for metastasis [147]. However, this metastatic cascade is an inherently inefficient process, prompting tumor cells to develop adaptations to increase their possibilities of survival. In this section, we focus on brain and CNS metastases and review the key steps and players involved in metastatic progression. Figure 2 offers a visual summary of this metastatic process and the primary elements playing a role in it.

### 4.1. The Pre-Metastatic Niche

First of all, tumor cells confirm their destination and prepare it for their arrival before leaving the primary tumor [148]. This co-evolution of tumor and target-organ microenvironment forms the “pre-metastatic niche” (PMN) [149], and it involves the interaction between tumor-derived factors, tumor-recruited cells, and local stroma [150]. The specific modifications that occur at the PMN are needed to favor the survival of tumor cells once they arrive, and a wide array of molecules and cells participate in this process.

Inflammation plays a crucial role in cancer progression and migration [151]. Studies have shown that hematopoietic progenitor cells from the bone marrow migrate to pre-metastatic sites and form clusters prior to tumor cell arrival, facilitated by receptor–ligand interactions [152]. At these sites, they establish an inflammatory chemokine gradient that attracts additional bone marrow-derived cells and tumor cells to the PMN [152], a process further supported by increased VEGF-induced vascular density [153]. Many inflammatory chemokines have been identified to help in the recruitment of bone marrow-derived myeloid cells to PMN and favor metastasis in several organs via suppression of host immunity, angiogenesis, and remodeling of tissue [154]. In the brain, granulocyte-derived molecules like lipocalin-2 (LCN2) trigger inflammatory activation of astrocytes, which in turn recruit myeloid cells to the brain [155]. Glycoprotein nonmetastatic melanoma B (GPNMB) expressed by macrophages and microglia is also linked to neuroinflammation, as it is upregulated in microglia, producing higher levels of inflammatory cytokines, including interleukin-1β (IL-1β) and tumor necrosis factor-α (TNF-α) [156]. Increased expression of GPNMB reduces T-cell activation by interacting with syndecan-4, enabling melanoma cells to escape immune detection and destruction [157]. There is also a contrasting anti-inflammatory role for GPNMB when bound to CD44 in astrocytes, showing deduction of nitric oxide synthase, reactive oxygen species, nitric oxide, and IL-6, but its exact effect on BMs has not been elucidated [158].

Cells in the PMN can produce pro-inflammatory cytokines and other pro-tumoral factors, but these can also come from distant cells either in the primary tumor or the bone-marrow and reach their target organs through extracellular vesicles. Extracellular vesicles selectively fuse with resident cells at their intended destinations, displaying unique integrin expression profiles linked to their metastatic organotropism [159]. Extracellular vesicles, including exosomes, microvesicles, and oncosomes, are membrane-enclosed particles released by various cell types into the extracellular space [160]. Extracellular vesicles can cross the intact BBB via transcytosis, facilitated by the upregulation of the endocytic pathway in brain endothelial cells, making them efficient carriers for tumor-derived factors [161]. These extracellular vesicles-derived factors interact with the soluble matrix, influencing the formation of a PMN by activating and recruiting inflammatory and resident cells [162,163].

Ruan et al. demonstrated how brain cells are reprogrammed by breast cancer-derived extracellular vesicles carrying high levels of miR-199b-5p, which targets solute carrier transporters in astrocytes and neurons, leading to the retention of metabolites in the extracellular space for tumor cells to use [164]. Exosomes carrying miR-122 microRNA can suppress glucose uptake by non-tumor cells in the PMN, which increases nutrient availability for tumor cells [165]. Proteomic analysis of exosomes derived from brain metastatic cells showed increased expression of hyaluronan-binding protein (CEMIP) [166]. Endothelial and microglial cells internalizing CEMIP-positive exosomes triggered angiogenesis and inflammation in the perivascular niche, facilitating brain vascular remodeling and metastasis [166]. BMs and PMN formation primarily take place in the brain parenchyma, traditionally regarded as an immune-privileged site [167]. The metastatic brain tumor microenvironment exhibits a unique cellular and non-cellular composition even when compared with that of primary tumors such as glioblastoma [168]. The extracellular matrix, an integral component of the PMN, has distinct protein arrangements regulated both by tumor and stromal cells, with organ-specific features [169]. For example, the extracellular matrix in metastatic brain tumors shows a highly organized appearance, with thick, dense collagen bundles clustering the tumor cell regions inside well-defined borders [168]. A proteomic analysis of NSCLC BMs has also shown an increased expression of invasion-related molecules such as integrin-α7, integrin-β1, and syndecan-4 in the extracellular matrix [170]. Expression of certain extracellular matrix proteins such as tenascin-C in the brain restricts T-cell migration, decreasing their concentration in the PMN and their ability to lyse tumor cells [171], promoting an immunosuppressive landscape.

Long non-coding (lnc) RNA increases the release of CCL2 to recruit macrophages in breast cancer BMs [172], while tumor cell-derived ANXA1 promotes microglial migration [173]. Later, both macrophages and microglia are activated via the PI3K signaling pathway to act as metastasis promoters [174]. The interaction between melanoma cells and microglia supports BM progression through melanoma-derived IL-6, which enhances STAT3 phosphorylation and SOCS3 expression in microglia, thereby aiding melanoma cell survival [175]. Moreover, increased expression of MMP-3 in microglia facilitated melanoma cell growth [176]. Both brain-resident microglia and bone marrow-derived macrophages are present in brain tumors and are actively recruited to the tumor microenvironment [177]. Microglia are regarded as the main contributors to CNS-resident tumor-associated macrophages (TAMs) [178]. TAMs represent the predominant type of immune cells detected in BMs, irrespective of primary tumor origin [168]. Karimi et al. used mass cytometry to analyze the tumor microenvironment of 139 high-grade gliomas and 46 BMs, achieving single-cell spatial resolution of immune lineages and activation states [179]. They found that monocyte-derived TAMs constituted about 30.5% of the tumor microenvironment, compared to 9.2% for resident microglia [179]. Collectively, TAMs in the microenvironment of BMs do not fall into the classic polarization of M1 (inflammatory, IFNγ-activated) and M2 (immune-modulating, IL4-activated) phenotypes [180,181], and their phenotype can remain highly plastic in response to different signals and interactions within the microenvironment [182,183].

Both TAMs and tumor cells can interact with lymphocytes in the tumor microenvironment of BMs to produce defective T cell states [181,184]. In an integrated analysis of brain tumor microenvironments, CD4+ T cells showed a hyporesponsive, anergic phenotype, while CD8+ T cells exhibited an exhaustion signature such as the one seen in chronic activation [181]. The chemokine interferon-γ inducible protein 10 (Cxcl10) is upregulated in myeloid-derived microglia, and despite its role as a chemotactic factor for lymphocytes, it also attracts more CNS-myeloid cells carrying immunosuppressive proteins that reduce T cell activation and promotes tumor growth [183]. Multiomics analyses have shown that in lung cancer BMs, the separation of myeloid and lymphoid cells into specific compartments is driven by unique cytokine networks [185]. Other cells, such as NK cells, neutrophils, and T cells, can also be found in BMs microenvironments, with distinct densities depending on the primary tumor type [179,184]. For example, in melanoma BMs there is higher neutrophil infiltration and increased CD8+ T cells in the margins of the tumor [179,181]. CD8+ T cells and other leukocytes have also been found in the CSF, matching the immune microenvironment of BMs and proving that cell exchange occurs between these compartments [186].

PMN formation in the brain, therefore, relies on intricate and diverse processes involving factors derived from tumor cells and brain-resident cells. The combination of inflammation, angiogenesis, immunosuppression, and selective brain-tropism help model the PMN, subsequently promoting tumor cell invasion and growth.

### 4.2. Lymphatic Spread of Tumor Cells

Given the genetic heterogeneity observed between primary tumor cells and their metastatic counterparts, it is suggested that tumor cells can arrive at their destination both through direct hematogenous spread from the primary tumor and sequential progression from lymph nodes [187]. In solid tumors, intravasation into lymphatic vessels and lymph nodes is more common and usually precedes metastasis to the vascular system [188], while tumor cells reach the lymphatic vessels alone or form clusters [188,189]. It has been demonstrated that once in the lymph node parenchyma, tumor cells can directly invade the lymph node vessels to enter the blood circulation, bypassing the thoracic duct [190,191].

Initially, the release of inflammatory chemokines from tumor and inflammatory cells in the tumor microenvironment induces lymphangiogenesis and enhances lymphatic intravasation [187]. Tumor-derived cytokines, soluble factors, and extracellular vesicles can prime the stromal cells in lymph nodes to create a lymphatic metastatic niche similar to the PMN [192,193]. VEGF-C overexpression in tumor cells has been shown to induce hyperplasia of the peritumoral lymphatic vessels, enhancing flow rate and delivery to lymph nodes [194]. The transport to lymphatic lymph nodes is actively regulated by signaling pathways, including the SDF-1/CXCR4 axis [195], the CCL1–CCR8 interactions [196], and the VEGF-C-induced upregulation of CCR7-CCL21 signaling [197]. VEGF-C enhances immune tolerance in murine melanoma models by inducing the deletion of antigen-specific CD8(+) T cells through lymphatic endothelial cells [198]. Chronic IFN signaling during the initial anti-tumor response induces epigenetic rewiring of tumor cells in the lymph nodes, upregulating PD-L1 and promoting immune tolerance [199,200]. Together with increased MHC-I expression, tumor cells are able to evade NK cells and resist T cell-mediated cytotoxicity [199]. Cancer cells in the lymph nodes also exhibit elevated expression of MHC-II, increasing regulatory T cells while decreasing CD4(+) T cells [201].

Tumor cells in the lymph nodes also undergo metabolic adaptations. Metastasis-initiating cells show elevated expression of the fatty acid receptor CD36 and lipid metabolism genes. Studies have shown that palmitic acid or a high-fat diet can enhance CD36 expression, increasing the metastatic potential of tumor cells [202,203]. Genes involved in the fatty acid oxidation pathway are upregulated in lymph node-metastatic tumor cells through bile acid-induced activation of the yes-associated protein (YAP) [204]. Moreover, lipid metabolism is also important for tumor cells to overcome ferroptosis in the lymphatic environment since the hallmark of ferroptosis is the lethal accumulation of lipid peroxidation products [205]. In the lymph, however, there are low iron levels and high levels of oleic acid, glutathione, and other antioxidants, decreasing exposure to oxidative stress and making tumor cells more resistant to ferroptosis when they enter the blood [206].

The reprogramming that occurs in the lymph nodes ultimately leads to a survival and metastatic advantage for tumor cells. Phylogenetic studies of metastatic breast cancer tumors have discerned that the genetic heterogeneity between primary tumors and their distant metastases comes from either a monoclonal metastatic precursor that evolves outside the primary tumor or polyclonal precursors originated from the primary tumor from the start [21]. Phylogenetic studies for other types of cancer have yielded similar results [207,208]. Mohammed et al. discovered that tumor cells reaching the lymphatic vessels exhibit a gene and protein profile indicative of a hybrid epithelial/mesenchymal phenotype and stem cell-like characteristics, which contribute to their high metastatic potential [188]. In mouse melanoma models, key driver mutations such as *BRAF* alterations and changes in genes like *MET* or *CDKNA2* (either gain or loss) are observed to occur within lymph nodes [209]. Cells with driver mutations combined with a loss of tumor suppressor genes that would normally help eliminate cells with mutations can, therefore, undergo transformation freely toward the development of melanoma [99]. Chromatin modifier histone deacetylase 11 (HDAC11) serves as a dynamic epigenetic regulator in the lymph nodes microenvironment as demonstrated in breast cancer cells models, in which increased HDAC11 expression inhibits cell cycle suppressors E2F7 and E2F8, promoting tumorigenesis and growth in the lymph nodes while downregulation of HDAC11 upregulates RRM2, promoting migration and egress from lymph nodes to distant sites predominantly through the draining blood vessels of lymph nodes [210].

### 4.3. Hematogenous Spread of Tumor Cells

Tumor cells can also enter directly from the primary tumor into the bloodstream. A clone subpopulation of breast cancer cells reprogrammed to overexpress the proteins Serpine2 and Slpi showed vascular mimicry and efficient blood intravasation [211]. However, only a small percentage of tumor cells that enter the circulation will survive the environmental pressures of the bloodstream to successfully metastasize [212]. CTCs, or tumor microemboli, initially enter the bloodstream as single cells but rapidly aggregate into clusters. This clustering, occurring after detachment from the primary tumor, is facilitated by the cancer stem cell marker CD44 through intercellular CD44-CD44 homophilic interactions [213]. Stemness of tumor cells is induced by epithelial-to-mesenchymal TRANSITION, and it supports migration [214]. Both mesenchymal and epithelial tumor cells enter the bloodstream and contribute to CTC clusters, with evidence indicating a dynamic plasticity between epithelial and mesenchymal states [214,215]. Hydrodynamic shear stress in the systemic circulation promotes epithelial-to-mesenchymal transition in CTCs. This process is driven by the generation of reactive oxygen species and nitric oxide and the suppression of extracellular signal-regulated kinase and glycogen synthase kinase 3β signaling pathways [215].

CTC clusters are more resistant to cell death than individual CTCS, increasing their potential to metastasize [216]. Their larger size allows them to overcome fluid shear stress and collisions with other cells in the circulation, promoting margination to the endothelium wall, which increases their probability of arrest and adhesion to wall receptors [217]. Despite their larger size, CTC clusters can navigate capillary-sized vessels by rearranging them into single-file chains [218], enabling them to reach distant organs. Additionally, capillary beds with slower flow rates promote CTC arrest and enhance active cell adhesion [219]. CTCs associated with neutrophils showed a transcriptomic profile that supported cell cycle progression and cell migration [220]. In later cancer stages, neutrophils show increased immunosuppressive functions, suggesting they have a dynamic role in metastatic progression [221]. Inflamed neutrophils form aggregates with CTC in the intraluminal space, but once CTC arrest occurs in a vessel, they lose contact with the tumor cells. However, they remain in close proximity to the clusters and endothelium due to chemokine signaling mediated by self-secreted IL-8, tumor-derived CXCL-1, and the endothelial cell glycocalyx [222]. IL-8 also causes endothelial barrier disruption and extravasation of nearby tumor cells [222]. Neutrophil extracellular traps (NETs) are neutrophil-derived DNA webs released in inflammatory states to trap and kill pathogens, but they can also trap CTCs [223]. They capture CTCs via β1-integrin, which is upregulated in inflammation both in CTCs and NETs [224]. Studies have shown that the interaction between β2 integrin on neutrophils and ICAM-1 on melanoma cells facilitates the anchoring of melanoma cells to the vascular endothelium [225]. ICAM-1 on triple-negative breast cancer cells promotes tumor cell secretion of suPAR, a chemoattractant for neutrophils, and attaches to CD11b molecules on neutrophils to form CTC-neutrophil bonds [226].

Platelets also interact and travel along CTCs, and a bidirectional exchange of lipids, proteins, and RNA occurs between them [227]. Tumor cells can transfer mutant RNA into blood platelets to produce “tumor-educated platelets” [228]. Platelets efficiently transfer structural components to tumor cells through extracellular vesicles, internalization, or direct contact, effectively “educating” tumor cells in the process [229]. Platelet-derived TGFβ and direct contact with CTCs activate the TGFβ/Smad and NF-κB pathways in tumor cells, driving their transition to a mesenchymal-like phenotype [230]. Platelet-derived RGS18 promotes the expression of the immune checkpoint molecule HLA-E in CTCs. As a result, CTCs can escape NK-mediated immune surveillance and killing [231]. Direct contact with platelets can upregulate the inhibitory checkpoint molecule CD155 in CTCs and inhibit NK-cell cytotoxicity when CD155 is engaged with immune receptor TIGIT [232]. The cross-talk between CTCs and platelets, therefore, creates highly dynamic and aggressive phenotypes that help preserve CTCs integrity during their transit in the bloodstream, enhance invasiveness and proliferation, perpetuate epithelial-to-mesenchymal transition and stem-like phenotypes, and evade death [229,233]. Platelets also stimulate YAP1 dephosphorylation and its nuclear translocation in CTCs, triggering a pro-survival gene expression profile that prevents anoikis in detached conditions [234].

CTC clusters are often accompanied by myeloid-derived suppressor cells (MDSCs), a group of immature myeloid cells that promote both systemic and local immunosuppression, forming a protective barrier around CTCs to aid in metastasis [235]. Aside from their immunosuppressive roles, CTC-MDSC interactions increase the production of reactive oxygen species in MDSC, which induces pro-tumorigenic differentiation and proliferation of tumor cells by upregulation of Notch1 receptor expression and activation in CTCs [236]. There have been attempts to oppose the immunosuppressive aspects of MDSC by targeting therapies against them. Some drugs have been approved by the FDA, some are undergoing clinical trials, and others are being investigated in preclinical models. However, there is still no consensus for their use [237].

Macrophages can have a dual role in regard to immunity against CTCs. On the one hand, CD24, a cancer stemness marker, can be expressed in tumor cells and play a suppressive role in tumor immunity as a phagocytic inhibitor when bound to macrophages via Siglec-10 [238]. On the other hand, Zhang et al. reported macrophages that engulf apoptotic tumor cells integrate tumor DNA into their nuclei, transforming into tumor stem cells while maintaining macrophage surface markers, enabling them to evade immune detection [239]. Therefore, some macrophages can promote metastasis while others interfere with it. More recently, Fu et al. [240] discovered that microbiota from the primary tumor can be carried by CTCs as intracellular bacteria capable of reorganizing the actin cytoskeleton of tumor cells and enhancing resistance to mechanical stress [240].

Tumor cells thrive in hypoxic conditions due to metabolic reprogramming [241], and CTC clusters offer protection against the toxic oxygen concentrations in the bloodstream [242]. Hypoxic CTC clusters promote a cancer stem-like phenotype in CTCs [243] and the acquisition of a reactive oxygen species-resistant phenotype that enhances CTC survival upon reoxygenation [244]. Oxidative stress and hypoxia favor CTCs to develop resistance mechanisms against anoikis, a form of apoptosis induced upon cell detachment from their native environment [245]. Several other adaptations have been linked to anoikis resistance, such as increased epithelial-to-mesenchymal transition, change in integrins’ profiles, oncogene activation, and overexpression of key metabolic enzymes or receptors [245]. The inclusion of carcinoma-associated fibroblasts in CTC clusters provides a metastatic advantage to tumor cells [246], likely favoring anoikis resistance, transportation of nutrients, and epithelial-to-mesenchymal transition [247].

### 4.4. Vascular Cooption

Vascular cooption describes the mechanism by which metastatic cells preferentially grow alongside the outer surfaces of existing blood vessels. This strategy is present in more than 95% of early micrometastases within the CNS [248]. Adhesion to vessels depends upon tumor cell β1 integrins adhesion to the vascular basement membrane and [248] the establishment of microcolonies [248].

In some organs, the presence of mesenchymal stem cells acting as pericytes at the perivascular space of the PMN mediate the extravasation of tumor cells [249]. It has been proposed that pericytic mimicry or angiotropism is a process closely related to vascular cooption and can be seen in melanoma cells that metastasize to the brain [250]. CTC from lung and breast cancers that metastasize to the brain utilize the cell adhesion molecule L1CAM to move along capillaries. This movement involves the activation of YAP through interactions with β1 integrin and integrin-linked kinase (ILK) [251].

Plasmin suppresses vascular cooption by deactivating L1CAM, an axon guidance molecule utilized by metastatic cells to navigate along brain capillaries. However, serpins that inhibit plasminogen activators produced by cancer cells, including neuroserpin and serpin B2, prevent the generation of plasmin. This inhibition facilitates vascular cooption in BMs associated with lung cancer, breast cancer [48], and melanoma [250]. Serpins also protect cancer cells by inhibiting the plasmin-generated FasL death signal [48]. A lncRNA associated with breast cancer cells increased expression of ICAM1, which mediated vascular co-option by increasing tumor cells’ ability to stretch over brain capillaries and extravasate into the brain parenchyma [172].

Intravascular cell arrest in brain microvessels before extravasation has been demonstrated to create a focal hypoxic microenvironment in the PMN, leading to ischemic changes that upregulate vascular remodeling factors such as Angiopoietin-2 (Ang-2) and VEGF [252]. Ang-2 facilitates tumor cell colonization and transmigration in the PMN and later supports stable oxygen and nutrient supply for metastatic growth [252].

### 4.5. Blood–Brain Barrier Penetration

There is evidence that the BBB, known as the tightest endothelial barrier, can be modified by soluble factors secreted by tumor cells or dysregulation of the normal brain microenvironment. Tumor-derived heparinase, for instance, can degrade the basement membrane of the BBB, facilitating tumor cell invasion into the brain [253,254]. Additionally, the absence of normal astrocytes leads to the downregulation of the DHA transporter Mfsd2a, expressed by endothelial cells, causing disruption of the BBB [255].

There are also tumor-derived extracellular vesicles that can be taken up by endothelial cells to increase the permeability of the BBB. Lung cancer cells secrete exosomes mediated by the action of TGF-β carrying lncRNA that increases the expression of MMP-2 in brain microvascular endothelial cells [256]. MMP-2 destroys tight junctions between endothelial cells both in the lung and the brain, increasing vascular permeability, tumor cell migration, and invasion [256,257]. Extracellular vesicles from breast cancer carrying miR-181c promote BBB disruption by altering actin dynamics [258], while extracellular vesicles containing miR-105 target the tight junction protein ZO-1 in endothelial barriers, compromising their integrity [259].

Tumor cells must acquire specialized adaptations before brain colonization, some of which involve specific mediators for BBB crossing. For example, breast cancer cells express COX2, the EGFR ligand HBEGF, and the α2,6-sialyltransferase ST6GALNAC5, which facilitate their traversal across the BBB [43]. ST6GALNAC5, in particular, was found to be specifically expressed only in brain-tropic metastatic cells, enhancing cooption to endothelial cells [43]. COX2 has been linked to the upregulation of MMP-1, which can degrade components of the BBB such as Claudin and Occludin [260]. In metastatic breast cancer cells, Klotz et al. demonstrated that semaphorin 4D (SEMA4D) regulates tumor cells’ transmigration through the BBB [46]. When SEMA4D binds to its receptor Plexin-B1 (PLXNB1) in endothelial cells, it makes them switch to a proangiogenic phenotype [261]. This effect may also be enhanced by TAMs [262]. Inactivating PLXNB1 has shown a shift in the immune landscape of tumor microenvironments towards an antitumor response; however, angiogenesis is not affected since SEMA4D can bind to alternative receptors [263].

Endothelial cells in the tumor microenvironment of BMs exhibit elevated Ki67 levels and enhanced microvascular proliferation. In contrast, the proliferation is suppressed in the presence of CD8+ T cells. Additionally, the tight-junction protein claudin-5, essential for BBB integrity, is downregulated in cancer cells located near endothelial cells, especially within the cores of BMs. This supports the hypothesis that vascular co-option plays a role in BMs colonization in regions with compromised endothelial junctions [179]. Herrera et al. found that breast-to-brain metastasis cell lines were able to traverse an enhanced blood-cerebrospinal fluid barrier (BCSFB) while primary breast cancer cell lines could not [264]. These findings reflected two things: firstly, cells that have previously colonized the brain must have acquired critical mechanisms to allow them to traverse CNS barriers; secondly, the preferential migration of breast cancer cells through the BCSFB may indicate it is an often-overlooked potential point of entry for tumor cells [264]. Under normal conditions, the BCSFB in the choroid plexus exhibits greater permeability than the BBB due to its transport and secretory functions [265]. Chemotherapies such as paclitaxel and 5-Fluorouracil (5-FU), commonly used in breast cancer, have been demonstrated to increase brain-barrier permeability to tumor cells, especially through the BCSFB due to upregulation of MMP-9 leading to Claudin-6 downregulation in the choroid plexus cells [266]. MMP-9 activity in the choroid plexus cells also resulted in the release of Tau from breast cancer cells, which formed neurofibrillary tangles that further destabilized the BCSFB [266]. Studies also have revealed that patients with parenchymal brain metastatic lesions often exhibit tumor cells in the ipsilateral blood–cerebrospinal fluid barrier [266].

Leptomeningeal disease (LMD) occurs when tumor cells invade the leptomeningeal membrane and the CSF [267]. Intracranial tumor cells spread via three mechanisms: direct perivascular pathways from the brain parenchyma, hematogenous routes from the systemic circulation, or iatrogenic seeding [268]. Extracranial tumor cell dissemination to the CSF occurs via hematogenous spread from the systemic circulation, backward migration along cranial or spinal nerves, invasion from the bone marrow via vascular pathways in the dura or skull, dissemination through meningeal lymphatic vessels, or through iatrogenic implantation [268]. Tropism for the meninges involves specific histological, molecular, and genetic alterations in the primary tumor cells [269]. Once in the leptomeninges, tumor cells continue adapting to overcome the intrinsic microenvironmental challenges of the CSF, including inflammation and sparse micronutrients [270]. Cancer cells within the CSF increase their expression of LCN2 when stimulated by inflammatory cytokines produced by CSF macrophages [270]. Besides its role in activating astrocytes in the PMN [155], LCN2 can also function as an iron-binding molecule. TAMs in the tumor microenvironment can help deliver iron to tumor cells to promote growth [271]. The uptake of iron in the CSF by tumor cells outcompetes macrophages that need iron to generate reactive oxygen species, therefore impairing the respiratory burst and phagocytosis functions needed for tumor control [270]. Tumor cells located in the cerebrospinal fluid produce complement component 3 (C3), which activates the C3a receptors on the epithelial cells of the choroid plexus. This activation compromises the BCSFB, permitting plasma elements like amphiregulin to enter the CSF and support the growth of tumor cells [272].

### 4.6. Astrocytes in Progression of Brain Metastases

Astrocytes serve as important mediators of BMs, as they can promote neuroinflammation, immunosuppression, angiogenesis, chemotaxis, and tumor cell invasion. Metastatic lung cancer cells release factors, including macrophage migration inhibitory factor, IL-8, and plasminogen activator inhibitor-1 (PAI-1). These factors activate astrocytes, producing inflammatory cytokines such as IL-6, TNF-α, and IL-1β, promoting increased tumor cell proliferation [273]. Schwartz et al. demonstrated that melanoma-secreted factors activate astrocytes to upregulate the expression of inflammatory chemokines such as CCL2, CXCL10, and CCL7, instigating astrogliosis, neuroinflammation, and hyperpermeability of the BBB [274]. Astrocyte-secreted CXCL10 has been demonstrated to facilitate the migration of melanoma cells toward astrocytes. This effect is attributed to the elevated expression of CXCR3, the receptor for CXCL10, in melanoma cells with a propensity for brain tropism [275]. Similarly, CCL2 can promote transmigration and extravasation of cancer cells via the CCL2-CCR2 astrocyte–cancer cell axis [276]. COX2 expressed in breast cancer cells increases prostaglandins, activating astrocytes to secrete CCL7, promoting self-renewal of tumor-initiating cells [260]. Soluble factors from triple-negative breast cancer cells induced upregulation and activation of the NLRP3 inflammasome in peritumoral astrocytes, consequently increasing IL-1β release, inflammation, and proliferation of metastatic cells [277]. There is evidence that in metastatic triple-negative breast tumors, IL-1β enhances the adhesion of cancer and immune cells to the brain endothelium via upregulation of cell adhesion molecules such as ICAM-1, VCAM-1, and E-selectin [278]. Lung cancer cells produce protocadherin 7 (PCDH7), which facilitates the creation of connexin 43 (Cx43) gap junctions with astrocytes. These connections enable the transfer of the second messenger cGAMP from tumor cells to astrocytes, thereby activating the STING pathway. Activation of this pathway leads to the secretion of inflammatory chemokines such as IFNα and TNFα. These chemokines act as paracrine signals for tumor cells to activate pathways such as the STAT1 and NF-κB signaling pathways, promoting their own growth and chemoresistance [44].

Astrocytes promote immunosuppression by significantly increasing the levels of neuronal-specific cyclin-dependent kinase 5 (Cdk5). This elevated Cdk5 reduces both the expression and functionality of class I major histocompatibility complexes, thereby disrupting the antigen presentation pathway [279]. Furthermore, reactive astrocytes with a signal transducer and activator of transcription 3 (STAT3) activation modify the innate and acquired immune system responses in the metastatic microenvironment [280].

Following early infiltration of tumor cells to the brain, activated astrocytes produce factors such as MMP-9, which promotes angiogenesis and release of growth factors from the extracellular matrix [281]. These signals persist as long as the astrocyte–tumor cell mutual association remains. Astrocytes also epigenetically upregulate Reelin expression in Her2+ breast cancer cells that migrate to the neural niche, conferring them a survival advantage in the brain microenvironment [282]. Peroxisome proliferator-activated receptor γ (PPARγ) in metastatic tumor cells activates astrocytes in the lipid-rich environment around the glial cells, enhancing cell proliferation in advanced BMs but not during early steps [283]. Astrocytes-derived exosomes containing *PTEN*-targeting microRNAs downregulate *PTEN* mRNA and protein expression in brain-tropic metastatic tumor cells [36]. PTEN loss in tumor cells facilitates perivascular brain colonization and invasion [284] and later increases secretion of the chemokine CCL2, which attracts myeloid cells, furthering metastatic proliferation [36].

## 5. Challenges Targeting Brain Metastases and Future Therapeutic Implications

Treatment for BMs includes neurosurgical resection, radiotherapy (i.e., either stereotactic radiosurgery, or whole-brain radiotherapy), and tumor-specific chemotherapy and targeted therapies [285]. The optimal therapeutic approach for each etiology of BMs will depend on the specific molecular and genetic landscape of the primary tumor. For example, HER2-positive patients can be treated with monoclonal antibodies such as trastuzumab or pertuzumab, whereas triple-negative breast cancer BMs treatment relies on BBB-permeable chemotherapeutics, such as capecitabine, cisplatin, and temozolomide [285,286]. In NSCLC BMs, tyrosine kinase inhibitors (erlotinib, gefitinib) have demonstrated good BBB penetration, and ALK inhibitors and ICIs are also available options for treatment [287]. In melanoma, chemotherapeutic agents have limited efficacy, which has started the investigation of a combination of immunotherapy and targeted therapy [288].

However, the development of resistance mechanisms limits the success of targeted therapies. Traditionally, resistance mechanisms were classified as intrinsic or acquired, but evidence has shown that the pattern of “acquired” resistance might be intra-tumor heterogeneity present from the start and expanded under selective pressures from targeted therapies [289]. Metastases and BMs, in particular, pose an extra challenge in determining targetable patterns and elucidating the development of resistance mechanisms. Over half of BMs harbor genomic alterations not found in primary tumors [19], prompting the need for direct BMs biopsies, which are not always accessible or feasible due to poor patient conditions [290]. Liquid biopsies have emerged as alternatives to analyzing tumor tissue. CTCs, cell-free tumor DNA (ctDNA), and extracellular vesicles from plasma and CSF can be collected and processed. However, most published research has been retrospective and performed in small, heterogeneous patient cohorts, and methodologic techniques for processing are diverse [291,292,293]. Translation into clinical practice must first overcome several technical barriers, including assay optimization and standardization, incorporation of liquid biopsies in more clinical trials, and creation of data biobanks to facilitate translational research [294].

The BBB represents a physical obstacle for chemotherapeutic agents to enter the brain. The integrity of the BBB varies in brain tumors and selectively excludes molecules based on factors such as electric charge, lipid solubility, and molecular weight. Disruption of the BBB using hyperosmolar mannitol has been investigated as a method to improve the delivery of large molecules, including proteins, antibodies, immunoconjugates, and viral vectors [295]. Low-intensity pulsed ultrasound with systemic microbubbles can increase BBB permeability as demonstrated with several agents in primary brain tumors [296]. Enhanced drug delivery with nanotechnology and nanocarriers has also been extensively researched [297]. Unfortunately, none of these methods has reached clinical feasibility yet.

Resistance in melanoma BMs with BRAFV600 mutations is primarily mediated by drug efflux transporters, including P-glycoprotein (P-gp; ABCB1) and breast cancer resistance protein (BCRP; ABCG2), located at the BBB and impeding penetration into BMs [298]. Moreover, the brain microenvironment may also exert an effect on resistance mechanisms that have not yet been fully elucidated.

Immunotherapies have become a pillar of cancer therapies, and ICIs targeting the PD-1/PD-L1 pathway, such as pembrolizumab (Keytruda), nivolumab (Opdivo), and atezolizumab (Tecentriq), have been approved by the FDA to treat different primary tumors presenting with BMs [299]. However, some patients exhibit resistance primarily due to the tumor microenvironment mechanisms such as defects in antigen presentation, cytokines signaling, presence of immune inhibitory molecules, and T cell exclusion [300].

There is an urgent need to advance BMs therapies and improve outcomes for patients. The gap in knowledge about mechanisms for metastases to the brain is still wide, and researchers must account for tumor heterogeneity and fast evolution when developing new therapies. Identifying more patients at risk or in the early stages of metastasis could further help researchers understand how different BMs develop and how to block their progression.

## 6. Conclusions

Brain metastases continue to be frequent causes of central nervous system tumors, and research has actively tried to determine the mechanisms that promote their formation. Advances in genetic and molecular sciences have allowed us to define models of the complex interactions between tumor cells, brain microenvironment, and host adaptations. However, there is still a gap in our knowledge about the molecular underpinnings of these tumors, what makes certain cells brain-tropic, and what makes them more equipped for survival in the hostile brain microenvironment. The heterogeneity of the tumors of origin and the infinite numbers of adaptations tumor cells can go through under different environmental pressures make this a highly dynamic field of study, which hinders the clinical applicability of study results for patients in a real-life setting.

We are reaching an era in which molecular studies in medicine require the intervention of other fields of research, such as artificial intelligence, machine learning, and computational simulation systems, which would allow for the processing of greater loads of information and hopefully create predictive models that help to determine tumor behavior, patient prognosis, and response to therapy using easily accessible samples (i.e., tumor circulating cells in the blood and primary tumor tissue), which remains a challenge for brain tumors.

## Figures and Tables

**Figure 1 ijms-26-02307-f001:**
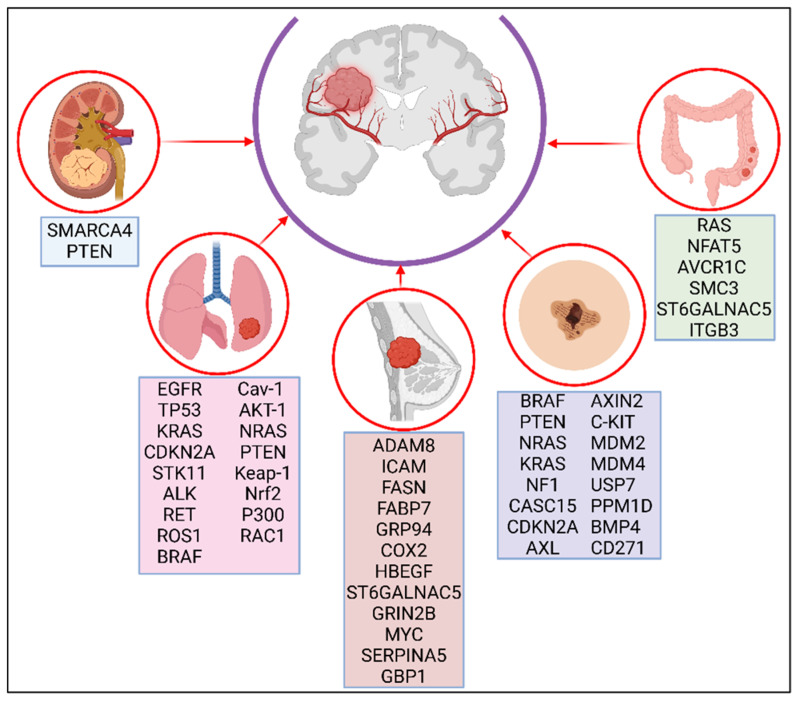
Genetic features of brain metastases in different cancer types.

**Figure 2 ijms-26-02307-f002:**
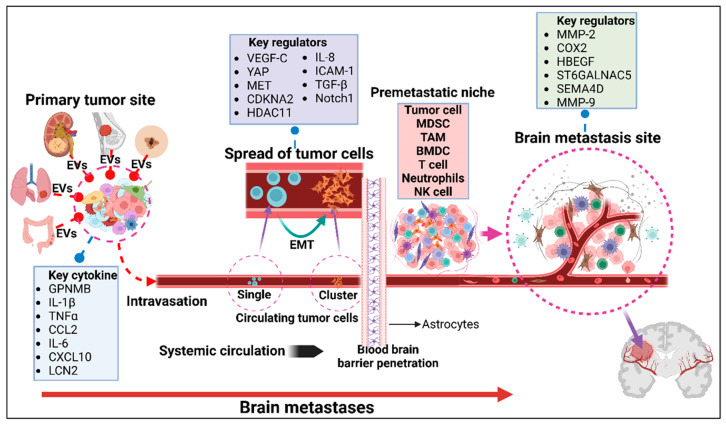
Molecular mechanisms of brain metastatic progression.

## Data Availability

Not applicable.

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
