# Peer review of "Molecular Underpinnings of Brain Metastases"

_ijms, 2025, doi:10.3390/ijms26052307_

Round 1
Reviewer 1 Report
Comments and Suggestions for Authors
See attached review

Author Response
Response to Reviewer 1
Opening comment: “In this submission to IJMS, the authors summarize current knowledge on the pathophysiology of brain metastases, from specific genetic characteristics of commonly metastatic tumors to the molecular and cellular mechanisms involved in progression to the central nervous system. The authors also briefly discuss current challenges in targeted therapies for brain metastases and explain the gap in knowledge that needs to be overcome to improve patient outcomes”
Response: Thank you kindly for taking the time to read and comment our manuscript.
Comment 1: I find this manuscript to be of interest to researchers specializing in molecular mechanisms in biological research as well as readers of this journal. As such, I am somewhat supportive of publication with some required edits. Specifically, this study attempts to focus on molecular mechanisms; however, some of the most molecular‐level investigations have used atomistic simulations to uncover these mechanisms, which should be noted: Biomater. Res. 2023, 27, 39 and J. Chem. Theory Comput. 2019, 15, 2807–2815. In particular, these prior approaches used atomistic simulations to understand molecular mechanisms in biological systems. Of course, I am not asking the authors to carry out additional calculations in this direction, but these prior studies should be noted as other related studies on molecular mechanisms. With this edit, I would be willing to re‐review this manuscript for subsequent publication.
Response: We appreciate the reviewer for this input. We decided to incorporate a line (line 895) on computational simulation systems in our newly written Conclusions section, as being one of the fundamental disciplines from which molecular studies in medicine can benefit from in future research.
Reviewer 2 Report
Comments and Suggestions for Authors
The manuscript was well-prepared with essential information about for-understand melanoma, metastasis, etc. The manuscript is accepted after minor corrections.
- The introduction is too short; include more information about words contained in the manuscript title
- Lines 131-134 of the Manuscript have some information that needs to be explained about “2 (HER2/neu), (ER-positive, PR variable, HER2-negative), luminal B (ER-positive, PR variable, HER2-positive or negative), HER2-enriched (ER-negative, PR-negative, HER2-positive), and basal-like (ER-negative, PR-negative, HER2-negative”, try to explain in short sentence about this codes or acronyms
- It is possible to include information by population, such as age, sex, ethnicity, Hispanic, Anglos, Africans, etc.
- The manuscript has several acronyms, and some this was difficult to understand
- Include information about the presence of cancer by genetically or habits
- Try to include information with references; it is recommended to remove tumors or chemotherapy
- The manuscript should include a conclusion
- Try to include more figures for the best understand
- It is possible to include countries with a population with a cancer problem
-The manuscript has several references (330); try to include only essential references
Author Response
Response to Reviewer 2
Opening comment: “The manuscript was well-prepared with essential information about for-understand melanoma, metastasis, etc. The manuscript is accepted after minor corrections.
Response: We appreciate the reviewer for taking the time to provide these constructive comments which we tried to address as best as we could to improve our manuscript.
Comment 1: The introduction is too short; include more information about words contained in the manuscript title.
Response: We appreciate the suggestion and added a couple of sentences to the introduction to serve as a link between the title and the content of our manuscript.
Comment 2: Lines 131-134 of the Manuscript have some information that needs to be explained about “2 (HER2/neu), (ER-positive, PR variable, HER2-negative), luminal B (ER-positive, PR variable, HER2-positive or negative), HER2-enriched (ER-negative, PR-negative, HER2-positive), and basal-like (ER-negative, PR-negative, HER2-negative”, try to explain in short sentence about this codes or acronyms.
Response: We appreciate the reviewer for the comment and apologize for the confusion. We subtly modified the lines immediately preceding the ones mentioned (now 131-133) in the hopes of making it clearer that depending on the levels of expression of the receptors, breast cancer can be subdivided in four subtypes. These same lines explain the meaning of these acronyms as these are then going to be used repetitively throughout the text.
Comment 3: It is possible to include information by population, such as age, sex, ethnicity, Hispanic, Anglos, Africans, etc.
Response: This is a thoughtful suggestion. However, we briefly touched on this in the epidemiology section of our manuscript, not wanting to delve deeper in this subject which falls more into the realm of epidemiology and not as much into the molecular sciences which is what we are trying to focus on with this review. We modified lines 68-70 to redirect readers to the work of a group focused on this subject.
Comment 4: The manuscript has several acronyms, and some this was difficult to understand.
Response: We agree with the comment and appreciate the suggestion. We’ve reduced the number of abbreviations utilized in the text in cases we considered appropriate (extracellular matrix, epithelial-to-mesenchymal transition, extracellular vesicles, reactive oxygen species). We also think some molecular names are too long and not using an abbreviation would hinder the reading of the text, so we’ve decided to keep them. Some words that are repeated multiple times (BMs for brain metastases, BBB for blood-brain barrier, etc) were also left as abbreviations to simplify reading.
Comment 5: Include information about the presence of cancer by genetically or habits.
Response: We appreciate the comment but think the point is not clear. The manuscript contains a full section on the genetic make up of the most frequent primary tumors that metastasize to the brain. In regards to habits of patients with brain metastases we welcome the reading of published studies in this subject but still would object that is not appropriate to include given the scope of our review is the molecular underpinnings of the tumors.
Comment 6: Try to include information with references; it is recommended to remove tumors or chemotherapy.
Response: We appreciate the comment but we are confused about the suggestion made in regards to the references, as all information has been dutifully referenced. As per the second part of the comment, we modified certain lines in the section on chemotherapies to be more concise and focus especially in the gaps that are yet to be overcome, which was the intention of that section. We’ll be open to further modifications if deemed necessary.
Comment 7: The manuscript should include a conclusion.
Response: We agree with the reviewer and decided to add a few lines to appropriately summarize and give our perspective in regards of where molecular research on brain metastases and brain tumors should be directed towards.
Comment 8: Try to include more figures for the best understand.
Response: We agree with the reviewer in regards of how figures can help with understanding complex texts. However, we already have incorporated 2 figures to the manuscript and we believe more than that would not really provide a better understanding.
Comment 9: It is possible to include countries with a population with a cancer problem.
Response: We appreciate the suggestion but as we previously discussed in our response to comment 3, such a discussion will be better addressed in an epidemiological study.
Comment 10: The manuscript has several references (330); try to include only essential references.
Response: We appreciate the suggestion and acknowledge the great number of references included. Trying to comply to the suggestion we’ve narrowed down the number of references to (302) keeping essential references and deleting some lines that were considered repetitive on second reading without compromising the overall quality of the text and the information being conveyed.